# Ovarian Morphometric and Histologic Characteristics and Correlation with Clinical Factors: A Cross-Sectional Study

**DOI:** 10.3390/jpm13020232

**Published:** 2023-01-28

**Authors:** Eirini Giovannopoulou, Maria-Valeria Karakasi, Maria Kouroupi, Argyro-Ioanna Ieronimaki, Eleni Papakonstantinou, Alexandra Giatromanolaki, Panagiotis Tsikouras, Pavlos Pavlidis

**Affiliations:** 1Laboratory of Forensic Sciences, School of Medicine, Democritus University of Thrace, 68100 Alexandroupolis, Greece; 2Department of Pathology, School of Medicine, Democritus University of Thrace, 68100 Alexandroupolis, Greece; 3Department of Pathology, General Hospital of Elefsina “Thriasio,” 19600 Elefsina, Greece; 4Department of Obstetrics and Gynecology, School of Medicine, Democritus University of Thrace, 68100 Alexandroupolis, Greece

**Keywords:** obstetrics, fertility correlates, reproductive health, ovarian morphometry, ovarian histology

## Abstract

Reproductive lifespan is determined by the reserve of ovarian follicles; their quality and quality determine the fertility potential at a given point in time for a particular individual. Inter-individual variations related to morphometry, laterality, medical history, demographic characteristics and ethnicity may impact ovarian histology, which however, has not been extensively studied or documented. The present cross-sectional study aims to investigate the potential association of clinical factors (age, medical and obstetric history) with ovarian morphometry and histology in females of reproductive age in the local population. The sample included 31 specimens of whole human ovaries, obtained from surgical/autopsy procedures in reproductive-aged women, processed at the Pathology Department. Morphometric characteristics were assessed, including shape, color, length, width, thickness and gross ovarian pathology. Random samples of specific dimensions were histologically examined to determine follicular counts. The results were analyzed statistically in correlation to morphometric characteristics and medical history. The majority of the patients had oval-shaped ovaries (77.8% right; 92.3% left; *p* = 0.368) of whitish color (38.9% right; 46.2% left; *p* > 0.999). Right ovaries had significantly greater length, width and volume (*p*-values 0.018, 0.040 and 0.050, respectively). Thickness was equivalent, as well as follicular distribution of all classes. Age correlated inversely with ovarian volume and primordial/primary follicular count on histology. Women with a caesarian-section history yielded significantly lower primordial/primary follicular counts. As estimated by ovarian histology, macroscopic and clinical factors may be significantly associated with actual ovarian reserve.

## 1. Introduction

Human ovaries lie within the lesser pelvis, laterally to the uterus and inside a shallow depression of the lateral pelvic wall termed as the “ovarium fossa”. They serve as reserves for ovarian follicles and reproductive hormones, playing a crucial role in female fertility and endocrine function [1]. The dimensions of the ovaries vary greatly throughout the lifespan of a female and exhibit considerable inter-individual differences, approximately ranging between 3–5 cm × 1.5–3 cm × 0.6–1.5 cm (length × width × thickness), and weighing 5–8 g [2]. They are smooth and pinkish during the early reproductive life. However, subsequent cycles of ovulation– scarring—corpus luteum and corpus alba formation distort their initially smooth surface [2,3].

The exact duration of the female reproductive lifespan has received accelerated interest in the recent years [4,5]. Reproductive aging is a matter of significant concern, as the mean age of childbearing has had an upward trend during recent decades [6]. Maternal age is firmly associated with the remaining ovarian reserve and has a substantial impact on the specific Assisted Reproductive Technology (ART) protocols and outcomes [7]. ART has a broad application in the field of infertility, assembling 2.0% of live births in USA in 2018 [8].

Ovarian reserve is defined by oocyte quantitative and qualitative characteristics and depicts the fertility potential of a woman at a specific time [4,9]. The need for quantification of the ovarian reserve has emerged during this process, mostly in terms of defining the prognosis and optimizing the protocols for ART [7]. Ovarian reserve is assessed by several clinical tools including transvaginal ultrasound measuring antral follicle count and ovarian volume as well as laboratory tests, including the follicle stimulating hormone (FSH), estradiol E2, and anti-Müllerian hormone (AMH) [10,11].

It has been documented that ovarian volume is associated with actual ovarian reserve, counted as the number of non-growing follicles present in histopathology [12]. Previous studies have demonstrated that an ovarian volume of less than 3 cm^3^ is associated with an increased risk of cycle cancellation in in-vitro fertilization (IVF) despite aggressive stimulation protocols. It is, also, associated with a requirement for higher doses of gonadotropins, and lower oocyte yields [3,13], suggesting a potential association of macroscopic characteristics such as volume with the actual number of recruitable follicles. Nonetheless, it has been suggested by previous findings that laterality (right versus left side) may significantly affect the ovarian volume and the antral follicle count (AFC) [14].

Concerning ovarian volume, anatomical descriptions for human ovaries are mainly retrieved from classic anatomy and histology textbooks [15,16]. There are several studies in the literature, concerning the morphologic and histopathologic features of human ovaries, but they are highly heterogeneous in terms of methodology, studied parameters, population, ethnicity and other potential variables that may affect the results. Nevertheless, the largest and most comprehensive studies regarding ovarian volume were conducted by means of ultrasound scanning [17,18]. There are reasonable concerns about the potentially limiting role of ultrasound technology, as many studies date back several years. The accuracy of ultrasound measurements may not reflect current practice, as ultrasound technology has greatly advanced over the years. However, the initial findings are enhanced by more recent publications [19].

Taking into consideration the growing interest in reproductive medicine both in terms of assisted reproduction and fertility preservation techniques, it is extremely useful to investigate the potential association of ovarian morphology and morphometry in relation to histopathology, past medical history, demographic characteristics and laterality in specific populations. Within our scope is to comment on the morphometric and morphologic characteristics of human ovaries and their association with histopathologic and clinical features in our study population, using a method which is cost effective and easily reproducible.

This study is motivated by the fact that variations related to ethnicity and environmental factors may have an impact on ovarian morphology and histopathology, which has not been extensively studied or proven. The findings aim to build a database for future studies in this particular population.

## 2. Materials and Methods

The sample of the present study consists of human ovaries prospectively collected from surgical or cadaveric specimens during the time period from November 2016 to November 2021 at (i) the Department of Pathology of Democritus University of Thrace, (ii) the Laboratory of Forensic Sciences and Toxicology of Democritus University of Thrace in Alexandroupolis, Evros and (iii) the Department of Pathology of “Thriasio” General Hospital of Elefsina, Attica in Greece. The samples were embedded in formaldehyde solution 10% and analyzed at the Department of Pathology in Democritus University of Thrace (Alexandroupolis, Greece). The study is in compliance with the declaration of Helsinki for medical research and has been approved by the Ethics and Research Committees of the University Hospital of Alexandroupolis and the General Hospital of Elefsina “Thriasio” (under documentation codes ES1/08-02-2018 and 185/22-06-2020, respectively).

All of the specimens that were collected during this time interval in females of reproductive age were included in the present study. Since the existence of differences in the different sides (right and left) is controversial in the literature [20,21], we have included both right-sided and left-sided ovaries, when applicable, in order to (i) avoid bias and (ii) run a comparison between the different sides.

A total of 19 patients and 31 ovaries were analyzed. Morphological characteristics such as dimensions (length, width, thickness), shape and color were measured. Subsequently, the specimens underwent histopathologic analysis, and the number of follicles was evaluated. The samples were subjected to the following process.

### 2.1. Macroscopic Observation

Macroscopic observation, including the shape and the color, was followed by measurement of the dimensions (length, width, thickness) for each ovary with electronic calipers. The presence of any gross pathology, such as cysts, was also addressed. The ovarian volume was estimated according to the following formula:V (Volume) = a × b × c × 4π/3,
where V comprises the measured volume in cm^3^ or ml, and a, b, c comprise each of the three radii of the ellipsoid in cm. The value of π is equal to 3.14. As the actual diameters were measured, the formula was modified as follows:V = l × w × t × π/6,
where l: length, w: width, and t: thickness (Figure 1).

### 2.2. Histopathologic Evaluation

Each ovary was further processed to obtain a random ovarian tissue specimen of fixed dimensions of 3 mm × 3 mm × 5 mm, containing an area of 3 mm × 3 mm of ovarian cortex. The tissue specimens, after being processed with the appropriate solution of alcohol and xylol, were embedded into paraffin wax. Slices of 5 μm every 50 μm of tissue were obtained from a microtome until the tissue specimens were exhausted. Only histopathology slices for which the whole surface was intact were included in the final analysis. As a result, a total of 25 slices were obtained for each ovary. Previous studies have reported measurements of variable width for cortical tissue such as 5 μm or 6 μm [22], 7 μm [23], 6 μm, 10 μm [22] or even 50 μm [24,25]. Taking into consideration that the mean diameter of a primordial follicle is estimated at approximately 39.5 μm (±7.6 μm) with a maximum up to 49 μm [24], the histopathology slice width was predetermined at 5 μm to retain a high quality of tissue and facilitate microscopic evaluation. To avoid double measurements one single 5μm slice was extracted from each 50 μm of cortical width. Based on previous studies, the random comparison and microscopic evaluation of ovarian tissue is sufficiently accurate to draw safe conclusions, in so far as the comparison and not the actual absolute number of follicles is the investigated outcome [26,27]. As noted, a total of 25 slices available for microscopic evaluation were obtained for each ovary included in this study. The tissue was dyed with conventional hematoxylin-eosin stain and observed with an optical microscope Nikon Eclipse 50i with magnification × 10 HPF (high-power field) (Nikon Group Companies, Tokyo, Japan). Follicular counts were obtained in a total of 6 samples from each ovary, randomly selected as the 1st, 5th, 10th, 15th, 20th and 25th of each series. Subsequently, they were classified based on their characteristics, as demonstrated by Gougeon et al. [28], to primordial, primary, secondary, and tertiary follicles (Figure 2). The slides were observed by two independent researchers and the results were cross-checked. The findings were statistically analyzed to detect any significant association in follicular number with age, BMI, laterality, ovarian volume, height, width, thickness, previous gynecological and obstetric history.

### 2.3. Statistical Analysis

Quantitative variables were expressed as mean values (Standard Deviation) and as median (interquartile range), while qualitative variables were expressed as absolute and relative frequencies. The Mann–Whitney test was used for the comparison of continuous variables between two groups. Spearman’s correlation coefficient (rho) was used to explore the association of two continuous variables. Multiple linear regression analysis was used with the volume and the number of primordial/primary follicles as the dependent variables. The regression equation included terms for all patients’ demographic and clinical characteristics. In cases where volume was the dependent variable, the number of primary cells was also included in the model as an independent variable. Linear regression analysis was conducted after the logarithmic transformation of the dependent variables. All reported *p* values are two-tailed. Statistical significance was set at *p* < 0.05 and analyses were conducted using SPSS statistical software (version 22.0).

## 3. Results

The sample consisted of 31 ovaries from reproductive-aged women with a mean age of 43.3 years (SD = 7.5 years) and regular menstrual cycles. Their characteristics are presented in Table 1. Mean BMI was 26.0 kg/m^2^ (±3.7) and 63.2% were overweight. Smokers comprised 36.8% of the sample. A total of 77.8% had an obstetrical history of at least one vaginal labor and 27.8% of at least one caesarian section. Only one woman had a history of malignancy (breast cancer) and 15.8% had undergone a previous adnexal surgery.

The majority of the patients had oval shaped ovaries (77.8% right ovaries and 92.3% left ovaries, *p* = 0.368). As far as the ovarian color is concerned, it was similar in left and right ovaries (*p* > 0.999) and whitish in the majority of the cases (38.9% for the right and 46.2% for the left ovaries). Right ovaries had significantly greater length, width and volume, compared to left ovaries (Table 2). Thickness was similar in left and right ovaries, as well as the distribution of all classes of follicles.

Spearman’s correlation coefficients of ovarian volume with follicular counts are presented in Table 3. Ovarian volume was significantly positively associated with primordial, primary and secondary follicles.

Greater age was significantly associated with lower volume (Table 4). Women over 45 years old had significantly lower volume compared to younger women. Moreover, significantly greater ovarian volume was associated with benign adnexal pathology as a primary indication for surgery, with type of surgery confined to adnexectomy or where there was a macroscopic appearance of a cyst.

Greater age was significantly associated with less primordial/primary follicles (*p* = 0.004). Women over 45 years old had significantly less primordial/primary follicles compared to younger women(*p* = 0.019). Furthermore, significantly less primordial/primary follicles were found in ovaries from women who had undergone a caesarian section in the past (*p* = 0.033).

With multiple linear regression, greater ovarian volume was associated with higher primordial/primary follicular density. In contrast, greater age and having had a caesarian section in the past were significantly associated with lower primary follicular density. Greater age was associated with significantly reduced ovarian volume (Table 5).

## 4. Discussion

The present study aimed to investigate the macroscopic and histopathological characteristics of human ovaries, regarding follicular counts, as well as their association with clinical factors, by using a method that is easily applicable and reproducible in common clinical practice.

The dominant shape was oval and the dominant color whitish, with no differences concerning laterality. Our findings suggest that right ovaries had significantly greater length, width and volume, compared to left ovaries. However, thickness was similar in both sides.

In the present study, ovarian volume was negatively affected by increasing age and primordial and primary follicles were negatively affected by both age and previous caesarian deliveries. Ovarian volume was independent from BMI, smoking habits and past obstetrical history (vaginal, caesarian deliveries, abortions). Clinical factors such as BMI, smoking habits, indication and type of surgery, previous adnexal surgeries, presence of a macroscopically evident cyst and previous vaginal deliveries do not seem to significantly affect the total follicular counts. Regarding the correlation of morphometry to histopathologic findings, increased ovarian volume was significantly correlated with increased follicular counts on ovarian histopathology.

Our findings demonstrate variability compared to previous studies. According to the findings of Rani et al., the predominant shape of human ovaries is almond in women aged 16 to 55 years [29]. This is inconsistent with our findings, where the predominant shape is oval at both sides. The authors, also, reported a gradual increase in all ovarian dimensions in older women. However, comparison between different sides (right or left) was not conducted, neither were the findings correlated with histopathology.

Perven at al. conducted a post-mortem study of human ovaries in the specific ethnic group of Bangladeshi women and found that the weight of right ovaries was significantly greater, compared to the left side [30]. As expected, the ovarian weight was significantly lower in the subgroup of women aged over 45 years. Of note, the findings were not correlated with histopathology [30]. In a previous study conducted in the same population, the authors found that right ovaries were associated with significantly greater length, breadth and thickness, compared with the left side and the findings were consistent within different age groups [21].

Ahmed at al. investigated the morphometric characteristics of human ovaries again in Bangladeshi women, including the variables of length, width and thickness [31]. Concerning the different age subgroups, the findings suggested that ovarian length was significantly greater in the older age groups, in both sides. The average width and thickness were significantly higher in the older age groups, but only on the right side. Breadth and thickness were significantly smaller in females aged under 13 years, but no significant differences in women over or under 45 years of age were observed.

Pavlik et al. determined the association of ovarian volume measured by ultrasound with age, height and weight in a large cohort of 58,673 women. Notably, ovarian volume negatively correlated with age, but also with small height [17].

The findings described above suggest high variability of the measured parameters, concerning laterality, age, and probably ethnicity, as specific ethnic subgroups are separately investigated. One would expect that as ovaries atrophy with ageing their morphometric characteristics would decrease in older age groups; however, this finding is not consistent among different studies.

The bottom-line in any case is the estimation of ovarian reserve. Several clinical and laboratory tools have been used in clinical practice to provide an indirect estimate of ovarian reserve, such as AFC or AMH [32,33,34]. Korsholm et al. conducted a cross-sectional study in a large sample consisting of 1423 reproductive-aged women in order to investigate antral follicular count and ovarian volume, measured by transvaginal sonography [14]. The study included both healthy and infertile women. The key findings suggested that right ovaries are larger and included more antral follicles than the left. To adjust for significant confounders such as age and AMH, patients were subcategorized to quartiles based on their chronological or biological (based on AMH) age. The difference in volume was consistent in all subgroups, except for AFC, which was significantly higher in the right ovary, except for the lowest AMH and the highest age quartile.

Alserri et al. conducted a retrospective observational study using a large sample of 6617 ultrasound scans, to assess a potential association of AFC with laterality [35]. The findings supported that antral follicular counts are significantly higher in the right side, compared to the left. This association was prevalent both in patients with polycystic ovarian morphology (PCO) and controls, when ovarian reserve was considered normal based on AFC counts [35]. Interestingly, the significant difference was eliminated in the subgroup of women with low ovarian reserve, as classified by an AFC count between 1 and 9 [35]. This is in line with findings of Korsholm et al. [14].

Jokubkiene et al. investigated ovarian volume and the antral follicular counts by using 3D-transvaginal ultrasound in 213 women receiving combined oral contraceptives (COCs). Based on their findings, laterality had significant impact on measured parameters. The right ovary was associated with greater volume, more antral follicles and larger predominant follicle, compared to the contralateral side, in the age group of 20–29 years. In women aged 30–39 years, the differences, although present, did not reach statistical significance [36]. A tendency towards reduced ovarian volume and less antral follicles was observed within older subjects (30–39 years of age) under COCs. Ovarian volume and vascularization, as measured by Doppler indices, seemed to decrease with advancing age, but the difference did not reach statistical significance. In this particular study, if only one ovary included a follicle > 10 mm, this was considered as “dominant” and a comparison of the measured variables in dominant and non-dominant ovaries was carried out, with no distinctions concerning laterality. The number of such cases was limited (24 cases) to allow for sound conclusions. Ovaries containing follicles >10 mm (dominant) did not present significant differences in ovarian volume and follicular number between the different age groups. On the contrary, the “non-dominant” ovaries were associated with diminished ovarian volume and antral follicles in the older age group.

Jokubkiene et al. also conducted a cross-sectional study with prospectively collected data in reproductive-aged women between 20 to 39 years to evaluate the normal 3D-transvaginal ultrasound findings in terms of ovarian volume, antral follicular count, total follicular volume and vascular Doppler indices [37]. The current use of COCs was an exclusion criterion. The analysis included a total of 303 women and the results were analyzed separately when all follicles measured <10 mm or when only one follicle in one ovary exceeded 10 mm. The findings suggested that right ovaries had significantly higher volume, more follicles and increased vascularization, compared with the left side, in the first subgroup. Ovarian volume and total follicular counts seemed to substantially diminish with age in both sides. Notably, AFC was significantly associated with total ovarian volume with a positive correlation.

In the subgroup of women with an ovary containing a follicle > 10 mm, the dominant ovary was associated with significantly higher volume but no difference in total follicular counts were observed between the two sides [37]. The total number of follicles were negatively associated with advancing age, a finding consistent with the group with no follicles exceeding 10 mm. Interestingly, ovarian volume of the dominant ovary was not affected by age, in contrast to the nondominant ovary; however, no distinction according to laterality was made.

Several theories for this laterization have been proposed in the current literature. Theoretically, a per se increased pool of primordial follicles in the right ovary established during early fetal development cannot be excluded [14]. Other potential causes may demonstrate their effect through accelerated follicular loss and atresia in the left-sided ovaries [14]. The anatomical peculiarities in blood drainage between different sides have been, also, implicated, as causal factors [35,36]. The drainage of the left ovarian vein is directed to the left renal vein, in contrast with the right draining directly to the inferior vena cava [38,39,40]. However, significant anatomical variations may exist, especially in venous drainage [39]. There are specific peculiarities concerning laterality, reflected also on the anatomy of the testicular veins that are involved in the etiopathogenesis of the varicocele in males [38]. Firstly, the inferior vena cava has a greater diameter. Secondly, the left renal vein develops a higher intravascular pressure, mainly due to its “entrapment” between the aorta and its branch, the superior mesenteric artery [41]. A wide spectrum of clinical manifestations arising from this “nutcracker phenomenon” may be presented in both genders, especially during the second and third decades of life [42,43]. Thirdly, the insertion of the left ovarian vein to the left renal vein forms a 90-degree angle [38]. These factors predispose to elevated intravascular pressure in the left gonadal vein, dilation with subsequent valve malfunction. The sequelae of blood stasis are increased local temperature and relative hypoxia, contributing to testicular dysfunction and impaired spermatogenesis in males with varicoceles [44]. A potential similar effect on females expressed as decreased ovarian volume and follicular number in the left side remains hypothetical [35].

An additional concern is that ultrasound evaluation of the left ovaries may be more challenging due to their anatomic location in close proximity with the sigmoid, potentially contributing to differences in measured parameters in favor of the right side [36]. However, these technical parameters are not considered to be a great contributor to these background differences, between the right and left side. The findings of our study, also, support the existence of a potential true biologic variation between the different sides.

Accurate knowledge of factors that significantly affect ovarian histopathology as defined per counted follicular density has significant clinical implications in assisted reproduction and fertility preservation. Our study contributes to the current knowledge by investigating the macroscopic characteristics of human ovaries and relating them to histopathology, demographics and clinical parameters, associated with the previous medical and obstetrical history.

## 5. Conclusions

Macroscopic and clinical factors may be significantly associated with actual ovarian reserve, as estimated by ovarian histopathology. Age was significantly associated with decreased ovarian volume and decreased counted primordial and primary follicles on histopathology. Furthermore, significantly decreased primordial and primary follicles were found among women with a history of a caesarian section.

### Strengths and Limitations

The strength of our study is that macroscopic characteristics were measured in surgical or autopsy specimens and not indirectly by means of ultrasound. However, the volume still represents an estimate, as it was calculated by a mathematical formula and the ovaries are not perfect ellipsoids [45]. A degree of error is, therefore, expected, but it should interfere with measurements on both sides. Additionally, the follicular counts were estimated by histopathology through a standard procedure for sample handling and inspection. The results were double-checked by two independent investigators. Additionally, all patients included in the analysis had a history of at least one delivery and proven previous fertility, comprising a healthy and fertile sample. Concerning the limitations, the number of patients is limited due to the content of our protocol. Despite the statistical significance of the findings, due to the limited patients’ number and the small absolute differences in follicular counts, the extent of clinical significance is unknown and cannot be determined by the present study [35]. An additional limitation is that the specimens—in some cases—derived from human cadavers. Cadaveric ovarian tissue histopathology may potentially deviate from surgical specimen histopathology. Although this difference has been investigated for other tissue types [46], cadaveric ovarian tissue histopathology has not been previously validated or standardized, and thus, a degree of error may be attributed to the source of the specimens. Our sample, however, predominantly consisted of surgical specimens, and therefore, the contribution of cadaveric specimens to the final outcome is proportionally low. This study does not aim to establish causality, but to highlight potential associations and areas of interest for subsequent studies.

## Figures and Tables

**Figure 1 jpm-13-00232-f001:**
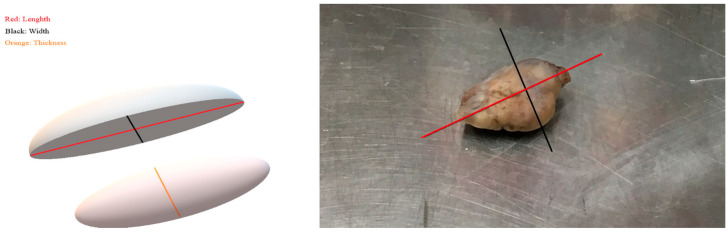
Graphical demonstration of the measured parameters. Macroscopic appearance of human ovary and demonstration of measured parameters; with red axis for length; black axis for width; orange axis for thickness.

**Figure 2 jpm-13-00232-f002:**
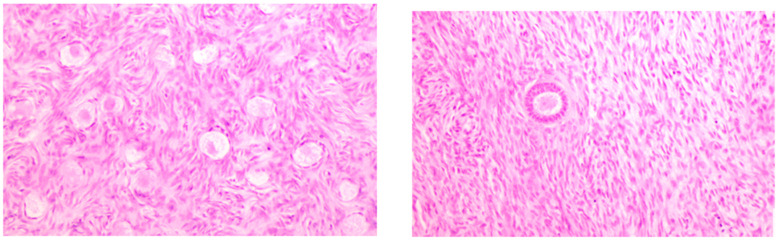
Different classes of counted follicles on ovarian tissue specimens. Observed in optical microscope Nicon Eclipse 50i with magnification × 20 HPF (High-power field).

**Table 1 jpm-13-00232-t001:** Sample characteristics.

Variables		N (%)
Age, mean (SD)		43.3 (7.5)
Age (years)		
	<45	9 (47.4)
	≥45	10 (52.6)
ΒΜΙ, mean (SD)		26.0 (3.7)
ΒΜΙ		
	Normal	5 (26.3)
	Overweight	12 (63.2)
	Obese	2 (10.5)
Smoking		
	No	10 (52.6)
	Yes	7 (36.8)
	Former	2 (10.5)
Vaginal deliveries		14 (77.8)
Caesarian deliveries		5 (27.8)
Spontaneous abortion		2 (11.1)
Induced abortion		5 (27.8)
History of malignancy		1 (5.3)
Previous adnexal surgeries		3 (15.8)
Macroscopically evident cyst		7 (36.8)
Primary indication for surgery benign adnexal pathology		
	No	13 (68.4)
	Yes	4 (21.1)
	N/A ^1^	2 (10.5)
Type of surgery confined to adnexectomy		
	No	15 (78.9)
	Yes	2 (10.5)
	N/A ^1^	2 (10.5)

^1^ cadaveric material.

**Table 2 jpm-13-00232-t002:** Measurements for each ovary.

	Right Ovary (Ν = 12, 94.7%)	Left Ovary (Ν = 12, 68.4%)	*p* ^+^		
	Mean (SD)	Median (IQR)	Mean (SD)	Median (IQR)	
Length(mm)	34.4 (12.2)	33.1 (29.5–35)	29.4 (5.9)	28 (24.5–33.5)	**0.018**
Width (mm)	23.1 (9.5)	20.5 (17.4–24.5)	19.1 (5.2)	17.5 (15.5–20)	**0.040**
Thickness (mm)	15 (13.2)	10 (8.5–16)	11.1 (3.7)	10.6 (8.5–13)	0.309
Volume (mL)	12.2 (28)	3 (2.5–7.6)	3.7 (2.6)	2.7 (1.7–4.6)	**0.050**
Primary follicles	6.3 (8.2)	4 (2–6.5)	7.3 (4.9)	7 (3.5–11)	0.099
Secondary follicles	1.7 (1.7)	1 (0.5–2)	2.6 (2.7)	2 (0–3.5)	0.198
Atretic	10.8 (7.7)	8.5 (6–14.5)	9.3 (6.9)	7.5 (4.5–12)	0.246

^+^ Mann–Whitney test.

**Table 3 jpm-13-00232-t003:** Spearman’s correlation coefficients of ovarian volume with the number of any type of cells.

Ovarian Follicles		Both Ovaries
Primary/primordial	rho	0.50
	*p*	**0.004**
Secondary	rho	0.37
	*p*	**0.046**
Atretic	rho	0.19
	*p*	0.297

**Table 4 jpm-13-00232-t004:** Association of volume with patients’ demographic and clinical characteristics.

		Volume (mL)		*p* Mann-Whitney Test
		Mean (SD)	Median (IQR)	
Age, rho		−0.45		**0.011**
Age (years)				
	<45	13.9 (26.6)	4.5 (3.1–10.1)	**0.045**
	≥45	4.6 (5.6)	2.6 (1.7–4.8)	
ΒΜΙ				
	Normal	8.3 (10.1)	3.1 (1.7–20.7)	0.887
	Overweight/Obese	8.5 (19.9)	3.4 (2.3–7.8)	
Smoking				
	No/Former	10.2 (22.2)	2.9 (2.3–6.6)	0.804
	Yes	5.3 (3.8)	4.1 (1.7–9.1)	
Primary indication for surgery benign adnexal pathology				
	No	4.5 (3)	3.1 (2.4–6)	**0.027**
	Yes	27 (37.2)	14.6 (4.5–25.1)	
Type of surgery confined to adnexectomy				
	No	8.4 (19.4)	3.8 (2.5– 7.1)	**0.033**
	Yes	22.9 (3.1)	22.9 (20.7–25.1)	
Vaginal deliveries				
	No	4 (2.5)	2.5 (2.2–6)	0.646
	Yes	10.4 (21.1)	3.9 (2–9.1)	
Caesarian deliveries				
	No	11 (21.5)	4.5 (2.4–9.1)	0.118
	Yes	3.2 (2.3)	2.5 (2–3.3)	
Spontaneous abortion				
	No	8.7 (19.3)	3.6 (2.2–8.5)	0.886
	Yes	10.1 (13)	3.8 (1.5–25.1)	
Induced abortion				
	No	10 (21.7)	2.7 (2– 6)	0.435
	Yes	5.8 (3.4)	5.9 (2.7–8.8)	
History of malignancy				
	No	8.7 (18.6)	3 (2.2–8.5)	0.469
	Yes	5.2 (1.2)	5.2 (4.4–6)	
Previous adnexal surgeries				
	No	9.4 (19.1)	4.1 (2.2–8.6)	0.112
	Yes	2.3 (0.9)	2.4 (1.7–2.8)	
Macroscopically evident cyst				
	No	3.9 (3.1)	2.6 (1.7– 5.2)	**0.013**
	Yes	16.9 (28.8)	4.8 (3.1–20.7)	

**Table 5 jpm-13-00232-t005:** Multivariate linear regression results with volume and number of primordial/primary follicles, in a stepwise method.

Dependent Variable	Independent Variable		β +	SE + +	*p*
Volume (mL)	Age		−0.03	0.01	**0.025**
	Macroscopically evident cyst	No (reference)			
		Yes	0.33	0.16	**0.047**
Primary follicles	Age		−0.05	0.01	**<0.001**
	Caesarian section	No (reference)			
		Yes	−0.42	0.15	**0.008**
	Volume (mL)		0.45	0.22	**0.050**

+ regression coefficient; + + Standard Error.

## Data Availability

Available upon request.

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
