# Peer review of "Ovarian Morphometric and Histologic Characteristics and Correlation with Clinical Factors: A Cross-Sectional Study"

_jpm, 2023, doi:10.3390/jpm13020232_

Round 1
Reviewer 1 Report
The authors find, by macro- and micro-anatomical investigations of 31 ovaries from 38 patients, statistically significant correlations: volume of right ovaries bigger than left ovaries, fewer follicles with increasing age and after caesarian section.
Form
Length of paper is appropriate in all parts. Only 7% of the references are from the last 5 years – this might be improved.
Language is good. Abbreviations in the legend of figure 2 and in the text “50i x 20 HPF” should be explained. Two references must be corrected for writing with only big characters.
Content
The paper contributes from the anatomical side to this still hot topic for clinical practice of ART: significance of parameters of ovarian reserve, like AFC, AMH and basal FSH.
Author Response
26-Jan-23
Prof. Dr. David Alan Rizzieri, Editor-in-Chief
Journal of Personalized Medicine
We would like to thank the Reviewers and the academic editor for taking the time to review our manuscript. Taking into consideration all the comments and recommendations, we have proceeded to a thorough review and revision of our manuscript. A revised manuscript is ready for resubmission, as encouraged. All the issues are addressed in detail, and we consider that this revision meets your recommendations and is eligible for resubmission. You can track the changes in the revised manuscript by using the relevant tool in MS word.
Point-by-point answers to suggested revisions
REVIEWER 1
Point 1. Length of paper is appropriate in all parts. Only 7% of the references are from the last 5 years – this might be improved.
Answer 1. We would like to thank the reviewer for the contribution towards improving our paper. The reference list has been updated. More recent references have been added. The percentage of references dated from 2017 and onwards is now 26%.
Point 2. Language is good. Abbreviations in the legend of figure 2 and in the text “50i x 20 HPF” should be explained.
Answer 2. We would like to thank the reviewer for the comment. The abbreviation of 50i refers to the type of the microscope used (microscope Nicon Eclipse 50i) and does not constitute a medical abbreviation. The abbreviation HPF refers to the magnification under which the picture was taken and stands for “High-power field”. The abbreviation is now explained in the text to facilitate understanding from the readers.
Point 3. Two references must be corrected for writing with only big characters.
Answer 3. We apologize for the inappropriate reference format. Appropriate changes have been made in the revised manuscript.
We are grateful and so honored to be working with you on this project.
I am looking forward to hearing from you
Yours sincerely,
G.E.
Reviewer 2 Report
This manuscript aims to describe the possible correlation between structural and histological features of ovaries with clinical data of patients. The final goal is to explore the concept of ovarian reserve from a different point of view.
The paper is well written in English and the whole comprehension of the purpose is guaranteed. However, the interest in the topic could be limited also by practical aspect and the results are based on a very restricted sample size. Even if the soundness and originality is not great, the paper has a value and could be improved by some revisions:
1. line 53. they could show more recent data than 2010
2. line 76. they should also take into account the limits of US in 2000
3. no reference in the discussion to the possible change due to cadaveric samples. Could this aspect affect the features of ovaries
Author Response
26-Jan-23
Prof. Dr. David Alan Rizzieri, Editor-in-Chief
Journal of Personalized Medicine
We would like to thank the Reviewers and the academic editor for taking the time to review our manuscript. Taking into consideration all the comments and recommendations, we have proceeded to a thorough review and revision of our manuscript. A revised manuscript is ready for resubmission, as encouraged. All the issues are addressed in detail, and we consider that this revision meets your recommendations and is eligible for resubmission. You can track the changes in the revised manuscript by using the relevant tool in MS word.
Point-by-point answers to suggested revisions
REVIEWER 2
Point 1. Line 53. they could show more recent data than 2010.
Answer 1: We thank the reviewer for the comment. The data are updated with more recent statistics, regarding the number of live births associated with the application of ART. The relevant reference has been replaced.
Point 2. line 76. they should also take into account the limits of US in 2000.
Answer 2. We would like to thank the reviewer for the remark. Indeed, the vast majority of evidence regarding ovarian volume has been accumulated before 2010, where the use of ultrasound may have played a restrictive role for the generalization of the results in current clinical practice. However, more recent studies present similar data and confirm the same trends1. The references have been updated in the revised manuscript. 2
Point 3. no reference in the discussion to the possible change due to cadaveric samples. Could this aspect affect the features of ovaries?
Answer 3: We thank the reviewer for this useful remark towards improving our paper. Despite evidence that human cadavers is a reliable source for histopathologic analysis3, the technique has not been standardized or validated specifically for ovarian tissue. Therefore, a deviation derived from cadaveric tissue cannot be excluded. However, the majority of our sample and observations are conducted on surgical specimens. In the section of limitations, we have added additional comments regarding this potential limitation.
We are grateful and so honored to be working with you on this project.
I am looking forward to hearing from you
Yours sincerely,
G.E.